# Adipose Tissue-Derived Stem Cells Retain Their Adipocyte Differentiation Potential in Three-Dimensional Hydrogels and Bioreactors [note 1]

**DOI:** 10.3390/biom10071070

**Published:** 2020-07-17

**Authors:** Benjamen T. O’Donnell, Sara Al-Ghadban, Clara J. Ives, Michael P. L’Ecuyer, Tia A. Monjure, Monica Romero-Lopez, Zhong Li, Stuart B. Goodman, Hang Lin, Rocky S. Tuan, Bruce A. Bunnell

**Affiliations:** 1Center for Stem Cell Research and Regenerative Medicine, Tulane University School of Medicine, New Orleans, LA 70112, USA; bodonne1@tulane.edu (B.T.O.); salghadban@tulane.edu (S.A.-G.); cives1@tulane.edu (C.J.I.); mlecuyer@tulane.edu (M.P.L.); tmonjure@tulane.edu (T.A.M.); 2Departments of Orthopaedic Surgery and Bioengineering, Stanford University School of Medicine, Stanford, CA 94305, USA; monik.rom@gmail.com (M.R.-L.); goodbone@stanford.edu (S.B.G.); 3Center for Cellular and Molecular Engineering, Department of Orthopaedic Surgery, University of Pittsburgh School of Medicine, Pittsburgh, PA 15213, USA; alanzhongli@pitt.edu (Z.L.); hal46@pitt.edu (H.L.); tuanr@cuhk.edu.hk (R.S.T.); 4Department of Pharmacology, Tulane University School of Medicine New Orleans, LA 70112, USA

**Keywords:** microphysiological system, tissue-on-a-chip, adipose stem cells, adipocytes, methacrylated gelatin, osteoarthritis

## Abstract

Osteoarthritis (OA) is a common joint disorder with a significant economic and healthcare impact. The knee joint is composed of cartilage and the adjoining bone, a synovial capsule, the infrapatellar fat pad (IPFP), and other connective tissues such as tendons and ligaments. Adipose tissue has recently been highlighted as a major contributor to OA through strong inflammation mediating effects. In this study, methacrylated gelatin (GelMA) constructs seeded with adipose tissue-derived mesenchymal stem cells (ASCs) and cultured in a 3D printed bioreactor were investigated for use in microphysiological systems to model adipose tissue in the knee joint. Four patient-derived ASC populations were seeded at a density of 20 million cells/mL in GelMA. Live/Dead and boron-dipyrromethene/4′,6-diamidino-2-phenylindole (BODIPY/DAPI) staining of cells within the constructs demonstrated robust cell viability after 28 days in a growth (control) medium, and robust cell viability and lipid accumulation in adipogenic differentiation medium. qPCR gene expression analysis and protein analysis demonstrated an upregulated expression of key adipogenesis-associated genes. Overall, these data indicate that ASCs retain their adipogenic potential when seeded within GelMA hydrogels and cultured within perfusion bioreactors, and thus can be used in a 3D organ-on-a-chip system to study the role of the IPFP in the pathobiology of the knee OA.

## 1. Introduction

Osteoarthritis (OA) of the knee affects more than 4% of people worldwide, making it the most common form of arthritis and a significant cause of physical disability, with significant medical and economic burdens. The percentage of the population with OA is predicted to increase as obesity and life expectancy are rising globally [1]. OA is characterized by the degradation of cartilage and the growth of bone spurs. OA is associated with many risk factors, including age, obesity, and trauma [2]. Current treatments of OA are limited to symptom management until severe cartilage degradation necessitates total knee replacement, demonstrating a critical need for a deeper understanding of OA pathologies.

Adipose tissue is a highly active endocrine organ, and its role as a contributor in the pathogenesis of many diseases, including OA, is beginning to be defined. Adipose tissue produces both pro-inflammatory cytokines and anti-inflammatory cytokines, demonstrating the complex role of adipose tissue in immune modulation [3,4,5,6]. The levels of several adipokines are increased in the synovial fluid of patients with OA, suggesting a potential link between adipose dysfunction and OA progression [7,8,9,10]. Additionally, magnetic resonance imaging and the analysis of blood serum and synovial fluid link synovitis and the inflammation of the local fat deposit, the infrapatellar fat pad (IPFP), with increased concentrations of inflammatory cytokines and OA severity/progression [11,12,13]. The dysregulation of cytokine production in adipose tissue, such as the increased levels of inflammation in the IPFP, of the knee in OA patients with obesity, has been reported and was positively correlated with OA pain [14,15,16,17,18]. While the metabolic disruption of adipose tissue is central in OA; it is important to note that adipose tissue is rarely included as a component of OA models [19].

Adipose tissues contain cells with mesenchymal stem cell (MSC) characteristics, a subset of which serve as progenitors for adipocytes, making them an ideal cell source for the creation of adipose tissue as they have nearly identical adipokine secretion profiles after adipogenic differentiation [20,21]. Adipose tissue-derived mesenchymal stem cell (ASC) adipogenesis is driven by the activation of peroxisome proliferator activated receptor γ (PPAR-γ) and is characterized by the accumulation of lipids and the induction of the expression of adiponectin (ADIPOQ), leptin (LEP), lipoprotein lipase (LPL), perilipin (PL1N), and fatty acid binding protein 4 (FABP4) [22,23,24,25]. ASCs isolated from different adipose depots have also shown variation in the differentiation potential and inflammatory properties [26,27]. Importantly, adipocytes differentiated in vitro from ASCs produce the cytokines considered important in OA [20,25,28]. Compared with subcutaneous ASCs and bone marrow-derived stem cells (BMSCs), IPFP-derived ASCs (IPFP-ASCs) have demonstrated increased chondrogenic differentiation potential [29,30]. Undifferentiated IPFP-ASCs have also displayed similar immunophenotype and adipogenic potential as ASCs derived from subcutaneous fat, however, the leptin expression varied [31]. Despite the extensive link between adipose and OA, it is unclear what portion of the adipose response observed in OA can be attributed to the IPFP.

Hydrogels based on collagen are widely used as biomaterials due to the abundance of collagen in the extracellular matrix (ECM) of native tissues, innate biocompatibility, and the ease of gelation [32]. However, the mechanical stiffness of native collagen hydrogels tends to be lower than required in tissue engineering, and rigid cross-linking often compromises cell viability. Gelatin, denatured collagen, retains many of the innate bioactivity properties of collagen, but it has a higher saturation density, which increases its stiffness to be more representative of native tissue. Previously, Lin et al. demonstrated that human BMSCs and induced pluripotent stem cells seeded on 15% (w/v) gelatin methacrylate (GelMA) effectively allows the modeling of the osteo/chondro interface of the knee [33,34]. To our knowledge, the differentiation of ASCs into mature adipocytes has not been studied in GelMA hydrogels in a perfusion bioreactor system.

The development of such a system is important for the construction of microphysiological systems (MPS) to simulate the degenerated articular during OA, as it will permit the recapitulation of the complex synovial microenvironment. By employing an MPS design, the interactions between adipose and other tissues can be investigated in the context of an environment that can be totally controlled. The data presented here demonstrate the generation of an engineered adipose tissue construct resulting from the adipogenic differentiation of ASCs in a GelMA-based 3D hydrogel, in the context of a custom-designed 3D printed bioreactor. The engineered adipose tissue has many uses including as a central component of an articular joint MPS our team is generating to model the pathogenesis of OA.

## 2. Materials and Methods

### 2.1. ASC Culture

Subcutaneous ASCs, passage 0 (p0), were purchased from Obatala Sciences (New Orleans, LA, USA). The cells were seeded at 400 cells/cm^2^ in 150 cm^2^ Nunclon plates (ThermoScientific, Pittsburgh, PA, USA) and expanded with growth medium consisting of Dulbecco’s modified Eagle medium Nutrient Mixture F-12 (Gibco, Gaithersburg, MD, USA), supplemented with 10% fetal bovine serum (Hyclone, Logan, UT, USA) and 1% antibiotic/antimycotic (anti/anti, Gibco) as previously described [35].

### 2.2. Hydrogel Synthesis

GelMA was synthesized as described previously [36]. Briefly, 15 g of gelatin (Sigma-Aldrich, Saint Louis, MO, USA) was dissolved in 500 mL of deionized H_2_O and then reacted with 15 mL of methacrylic anhydride (Sigma-Aldrich, Saint Louis, MO, USA) overnight at 37 °C. The solution was then dialyzed against water in a 3500 molecular weight cut off (MWCO) dialysis cassette (ThermoScientific) for 5 days and lyophilized in a ThermoSavant freeze-drier (ThermoScientific) for 3 days.

### 2.3. Cell Encapsulation and Bioreactor Set-Up

GelMA was dissolved at 15% (w/v) in Hank’s balanced salt solution (Gibco) containing 0.15% (w/v) lithium phenyl-2,4,6- trimethylbenzoylphosphinate (LAP, Sigma-Aldrich) and 1% anti/anti. ASC pellets (p. 3–4) were suspended in the GelMA solution at 20 × 10^6^ cells/mL and the gelation was photoactivated using a dental curing light (395 nm wavelength; LEDWholesalers, Hayward, CA, USA) for 2 min. Constructs were placed in static culture in NUNC 48-well plates (ThermoScientific) or as dynamic culture in a custom-designed 3D printed bioreactor [27]. Figure 1 illustrates the custom-designed dual stream 3D printed bioreactor. The media were perfused through the system at 5 μL/min using a programmable syringe pump (NewEra, Farmingdale, NY, USA). Conditioned medium samples were collected for future analysis.

### 2.4. Construct Culture and Adipogenesis

All the constructs were cultured in a growth medium or a commercially available adipogenesis medium (AdipoQual, Obatala Sciences, New Orleans, LA, USA). In static culture, the medium was replaced after 2–3 days, while the bioreactor medium was refilled as needed. After 28 days, the constructs were harvested for subsequent evaluation.

### 2.5. Live/Dead Staining

In order to determine the cell viability after 28 days of culture, the constructs were first washed in Dulbecco’s phosphate buffered saline (DPBS, Gibco) before being stained with a Live/Dead staining kit (ThermoScientific) according to the manufacturer’s protocol. Images were acquired on a Cytation 5 multi-mode reader using Gen5 imaging software (BioTek, Winooski, VT, USA). The area of positive calcein stain was determined in two fields per sample using a custom MATLAB code utilizing MATLAB R2020a software (Appendix A).

### 2.6. Alamar Blue Staining

Alamar blue staining is a common method for measuring cell proliferation. The alamarBlue™ Cell Viability Reagent was purchased from ThermoFisher and the staining was conducted according to the manufactures protocol with slight modifications. Briefly, the staining solution was diluted one to ten in growth media and static scaffolds were incubated overnight at 37 °C and 5% CO_2,_ completely submerged in staining solution. The static scaffolds were stained after 7, 14, 21, and 28 days of culture with growth media. Absorbance at 570 nm was measured using a Synergy multi-mode plate reader (BioTek).

### 2.7. Neutral Lipid Staining

To visualize the neutral lipid accumulation, two different methods were employed, Oil Red O and boron-dipyrromethene (BODIPY) staining. After the constructs were harvested, they were fixed in 4% paraformaldehyde (PFA) and stored at 4 °C overnight before staining.

Whole constructs were washed with PBS before being placed in a 0.5% filtered solution of Oil Red O (Sigma-Aldrich) for 1 h. The constructs were briefly washed in 60% isopropanol before being washed in a PBS solution until the washes were clear. Images were then acquired at 20× magnification on a Nikon Eclipse TE200 microscope equipped with Nikon Digital Camera DXM1200F and Nikon ACT-1 software version 2.7 (Nikon, Melville, NY, USA).

The fixed constructs were placed in a 30% sucrose solution overnight and then overnight in OCT cryoembedding medium (Sakura, Torrance, CA, USA), after which the constructs were washed twice in OCT for 45 min and then frozen at −80 °C.

BODIPY/4′,6-diamidino-2-phenylindole (DAPI) staining was performed on cryosections (14 μm) after heat fixing the slides at 55 °C for 30 min and washing the OCT from the slides with 4% PFA for 15 min. The sections were placed in 2 μM BODIPY (Invitrogen, Waltham, MA, USA) in PBS for 15 min at 37 °C. The slides were then mounted and coverslipped in VECTASHIELD Antifade Mounting Medium with DAPI (Vector Lab, Burlingame, CA, USA). The slides were imaged on Cytation 5 multi-mode reader (BioTek).

### 2.8. Quantitative Polymerase Chain Reaction (qPCR)

RNA extraction was done using RNeasy Microkit (Qiagen, Germantown, MD, USA) according to the manufacturer’s protocol. The reverse transcription to the complementary DNA (cDNA) of 1 μg of RNA was completed using a High-Capacity cDNA Reverse Transcription Kit with RNase Inhibitor (ThermoScientific).

The qPCR was performed with SsoAdvanced Universal SYBR Green Supermix (Bio-Rad, Hercules, CA, USA) with custom primers ordered from Integrated DNA Technologies (IDT, Newark, NJ, USA). The fold change in gene expression was calculated using the −∆∆C_t_ method with reference to glyceraldehyde 3-phosphate dehydrogenase (GAPDH) as a housekeeping gene and compared to the control growth medium groups in the monolayer, static, and dynamic cultures. Gene primer sequences are listed in Table 1.

### 2.9. Enzyme-Linked Immunosorbent Assay (ELISA)

Collected samples of the conditioned cell culture media were cleared by centrifugation at 1000× *g* for 1 min before being stored at −80 °C. The samples were brought to room temperature before analysis using two commercially available ELISA kits according to the manufacturers’ protocols. Human ADIPOQ ELISA kit (ThermoScientific) and LPL (Cell Biolabs, San Diego, CA) ELISA kits were used. Absorbance at 450 nm was measured using a Synergy multi-mode plate reader. The conditioned medium from the dynamic cultures was concentrated using a 10 kDA filter. The ELISA data were expressed as a function of the total protein content, which was determined using a Pierce BCA kit (ThermoScientific).

### 2.10. Statistical Analysis

One-way ANOVA with Tukey comparisons and Student t-tests were performed using GraphPad 8.3 (GraphPad). All graphs are the mean data ± standard error of the mean and all the experiments were performed in triplicate. Significance was determined as *p* < 0.05 and is denoted by *. If *p* < 0.01, the significance was denoted by **, *p* < 0.001 by ***, and *p* < 0.0001 by ****.

## 3. Results

### 3.1. Cell Viability and Morphology

After 28 days of culture, the ASCs cultured in growth medium retained their spindle morphology, while the ASCs cultured in the adipogenic medium became spherical with multilocular lipid droplets filling the majority of the cell volume. In static 3D culture, the ASCs cultured in either growth or adipogenic medium demonstrated robust cell viability after 28 days as indicated by a majority of cells being fluorescent positive for Calcein (Figure 2A,B). In dynamic 3D culture, a higher frequency of cell death was observed compared to static culture (Figure 2C,D). The quantification of calcein positive staining within the scaffolds confirmed a significant decrease in live cell staining between static and dynamic cultures (Figure 2E, *p* < 0.01). Additionally, cell death occurred most frequently along the edges of dynamic cultures and in small groupings distributed randomly throughout the constructs. Similar to the static culture, the ASCs seeded in dynamic constructs and cultured in a growth medium retained their spindle morphology, while the ASCs seeded on the dynamic constructs culture in adipogenic medium became spherical (Figure 2B,D). The proliferation of ASCs while cultured with growth media in static culture showed a trend to increase over a four-week time period and the standard deviation between the samples decreased (Figure 2F).

### 3.2. Increase in Lipid Accumulation in Cultures Maintained in Adipogenic Medium

ASCs grown in the adipogenic medium demonstrated differentiation as indicated by the cytoplasmic accumulation of droplets of neutral lipids detectable by both Oil Red O (Appendix AB,D) and BODIPY staining (Figure 3B,D,F,H), as compared to the ASCs seeded on static and dynamic 3D constructs cultured in growth medium (Figure 3A,C,E,G). Established destaining protocols for Oil Red O failed to remove the stain from the GelMA hydrogels, possibly due to the limited diffusion through the constructs. BODIPY staining in differentiated constructs was concentrated around the nucleus, and the cells demonstrated a spherical morphology (Figure 3B,D). Interestingly, the dynamic 3D constructs exhibited a cell colony formation, while the static constructs had an even distribution of cells.

### 3.3. Confirmation of Adipogenic Differentiation Based on Lineage-Specific Gene Expression

Adipogenic differentiation was further confirmed using a qRT-PCR analysis for the expression of the key adipogenic marker genes, *PPAR-γ*, *APN*, *PL1N*, *LPL*, and *FABP4*. Upon adipogenic induction, the expression level of *PPAR-γ* mRNA in differentiated ASCs was increased ~3000-fold in monolayer cultures, 19-fold in static 3D cultures, and 70-fold in dynamic 3D cultures as compared to their corresponding undifferentiated ASCs (Appendix AA). Similarly, the increased levels of the gene expression of *APN* (4.18 × 10^6^, 4.45 × 10^4^, 2.44 × 10^3^), *PL1N* (7.48 × 10^3^, 1.67 × 10^3^, 6.70 × 10^1^), *LPL* (1.92 × 10^6^, 4.67 × 10^3^, 5.16 × 10^2^), *FABP4* (2.84Ε × 10^6^, 9.24 × 10^3^, 9.01 × 10^3^) were detected in differentiated ASCs monolayer, static, and dynamic cultures, as compared to the corresponding undifferentiated ASCs (Table 2, Appendix AB–E). The change in the *LEP* gene expression was greater than 10-fold in the dynamic and static 3D cultures cultured in an adipogenic medium as compared to the growth medium controls; however, the expression was more variable in the 3D cultures compared to the monolayer cultures (Appendix A).

### 3.4. Production of Adipokines

The levels of ADIPOQ were significantly increased in the samples of the conditioned adipogenic medium collected from both the monolayer and the static 3D cultures as compared to those in the conditioned growth medium (Figure 4A, *p* < 0.0001). ADIPOQ levels were significantly decreased in the static 3D culture compared to the monolayer culture, and the concentration in the dynamic 3D cultures was decreased when compared to both the monolayer and static cultures (Figure 4A, *p* < 0.0001). The level of LPL in the media collected from the monolayer and static 3D cultures was increased in the conditioned adipogenic media as compared to the conditioned growth medium (Figure 4B, *p* < 0.05). For the medium collected from the dynamic 3D cultures, the filtration-mediated concentration was needed to achieve the detectable levels of LPL, and the results showed significantly higher levels of LPL in the adipogenic medium as compared to the growth medium (Figure 4C, *p* < 0.05). ADIPOQ and LPL protein concentrations were undetectable in the unconcentrated conditioned media collected from the dynamic cultures and the unconditioned adipogenic media. The low levels of tested molecules in the dynamic culture may be due to the rapid removal of secretary factors through microfluidics.

## 4. Discussion

This study aims to demonstrate that ASCs retain their adipogenic differentiation potential when encapsulated within photo-cross-linked GelMA hydrogels as a 3D construct and cultured within a perfusion bioreactor. We observed high cell viability and neutral lipid accumulation within the constructs cultured with adipogenic media. Mature adipocytes are characterized by one large lipid-filled intracellular vacuole that pushes all the organelles to the cortex of the cell. The neutral lipid staining demonstrated that the ASCs grown in the growth medium had only non-specific background staining, while the ASCs cultured in an adipogenic medium exhibited lipid accumulation that filled the cells. However, neutral lipid staining demonstrated that instead of one singular lipid-filled vacuole observed in vivo, there were several large lipid-filled vacuoles within each cell, consistent with previous observations that ASCs differentiated in vitro into mature adipocytes are filled with many lipid deposits [25,28]. Additionally, BODIPY staining demonstrated that lipid-filled intracellular vacuoles accumulated within the cell cytoplasm consistent with grade 4 differentiation on the scale published by Aldridge et al. [37]. Upon adipogenic induction, *PPAR-γ* gene expression was robustly increased in monolayer, static, and dynamic culture, which has previously been shown to be sufficient to initiate adipogenesis [25,38]. The adipogenic differentiation of the cells is further supported by the increased expression of mature adipocyte-specific genes, *ADIPOQ*, *FABP4*, *PL1N* and *LPL*, consistent with the previous findings on adipogenesis [39,40]. Finally, differentiated cells significantly increased the production of the mature adipocyte cytokines, ADIPOQ and LPL, in the monolayer and static cultures. The medium samples from the bioreactor cultures, after appropriate concentration, also demonstrated similar significant increases in LPL. The presence of these proteins is consistent with the previous literature describing the ASC differentiation into mature adipocytes [25,28,41]. The findings indicate the successful differentiation of ASCs towards mature adipocytes in both static and dynamic cultures. To our knowledge, this is the first time 3D engineered constructs consisting of cells encapsulated in GelMA hydrogels were used to model adipose tissue in both static cultures and dynamic cultures within a perfusion bioreactor.

Surprisingly, a decrease in the gene expression of both the adipogenic markers and the cytokines was noted in both the static and dynamic 3D cultures when compared to the monolayer cultures, which is potentially due to the complex diffusion-related challenges in terms of the delivery of pro-adipogenesis factors into the 3D hydrogels. While no necrotic core was observed during the live/dead imaging, the dead cells were concentrated along the edges and within small groups in the dynamic culture constructs. Fifteen percent (w/v) of the gelMA mechanical properties were thoroughly characterized by Van Den Blucke et al. [42] and Hang et al. [36]. Van Den Blucke et al. demonstrated a decreased porosity and increased fiber diameter in the methacrylated gelatin as compared to native scaffolds. The restricted pore size could limit the diffusion of medium through the scaffold. Additionally, protein–scaffold interactions are as of yet unstudied, potentially contributing to the sequestration of pro-adipogenesis factors. Furthermore, no published work has investigated the microstructural organization of GelMA. Concentrations of gelatin fibers may lead to unforeseen protein interactions and complicate the diffusion of media through the constructs. Both issues may be compounded in dynamic cultures where increased medium volumes further dilute the adipokines produced by the cells, and is coupled with a limited diffusion interface. This could account for the requirement for the concentration of the conditioned medium collected from dynamic cultures to observe the detectable differences between the growth medium and adipogenic medium cultures.

LEP in the serum is a commonly highlighted adipokine in obesity-related OA. Hui et al. demonstrated that the exposure to LEP leads primary human-derived chondrocytes to reduce the expression of collagen type II while increasing the level of the expression of *MMP-1* and *MMP-13* [9]. Additionally, Dumond et al. demonstrated increased LEP concentrations in the synovial fluid of OA patients. In contrast, Ding et al. demonstrated that LEP concentration in synovial fluid was negatively correlated with cartilage thickness [7,43]. In this study, whereas monolayer cultures demonstrated the expected increase in *LEP* expression compared to undifferentiated controls, static and dynamic 3D constructs both showed varying increased and decreased *LEP* expression. We have previously noted that *LEP* expression is affected by the components present in the adipogenic medium. The use of a commercially available adipogenic medium in these studies makes it challenging to assess the effects of the individual components on this variation as the specific constituents are proprietary. However, this will be investigated by using several adipogenic media of different compositions (Appendix A).

As noted, it is unclear as to how the interactions with the scaffold, due to non-specific interactions or the trapping of proteins, can affect the diffusion of adipokines into the medium. The ADIPOQ and LPL ELISA data demonstrated that a portion of the adipokines did diffuse out of the scaffolds into the medium streams, but it is likely that a portion of the adipokines remain trapped in the scaffold. Trapped adipokines combined with the increased amount of medium required for a perfusion system contributed to the need to concentrate the conditioned medium to achieve detectable concentrations for analysis. Additionally, the studies have demonstrated that the homeostasis of cartilage, in terms of the expression of collagen type II and aggrecan, is susceptible to mechanical stimulation [44]. For the current bioreactor design, the major purpose is to establish a methodology for future studies to investigate the underlying molecular interaction(s) between adipose tissue and immune components of the knee. Future iterations will permit the potential incorporation of mechanical stimulation into our model to better mimic in vivo physiology.

Future studies will utilize this established model to investigate the interactions between adipose tissue and other tissue components of the synovial joint. The validation of the model will establish a baseline for future studies in which the perfusion bioreactor used here will be placed in series with bioreactors containing other articular joint tissues to create a micro-joint MPS. An OA-like pathology will be created in the MPS and the effectiveness of the potential therapeutics may be tested. The prevalence of knee OA underscores the need to develop an MPS that simulates the anatomical and physiological characteristics of the knee. This study has demonstrated that the ASCs isolated from fat depots could be used to fabricate an engineered construct that models adipose tissue. IPFP-ASCs have been highlighted as a potential therapeutic for patients with injured knees [29,30,45]. In addition to their well characterized chondrogenic potential, IPFP-ASCs have therapeutic potential to manage substance P (SP), expressing neuronal ingrowth into the synovium and the IPFP. Kouroupis et al. demonstrated that under regulatory compliant conditions, IPFP-ASCs express CD10 surface markers, thought to be responsible in degrading substance P-expressing neurons [46,47]. SP neurons are involved in nociception and inflammation management, and SP neuron ingrowth into cartilage, synovium, and the IPFP are hallmarks of OA and synovitis [48]. Through the degradation of SP neurons, IPFP-ASCs could be used to manage pain and work in multiple ways to decrease inflammation. Previous literature suggests that obesity and type II diabetes mellitus alter ASC therapeutic properties; however, there are no studies as of yet that have investigated if these changes are found in IPFP-ASCs [49,50]. Therefore, IPFP-ASCs will be used to engineer adipose tissue for the knee-specific MPS. Furthermore, the synovium is often highlighted as the largest source of immune cells, particularly macrophages, that invade the synovial capsule in patients with OA and drive cartilage degradation [51]. Klein-Wieringa et al. demonstrated that the IPFP has a similar composition of immune cells to that of the synovium in patients with OA [52]. The adjacency of IPFP to the synovium suggests that the IPFP may also be a source of disease-altering immune cells. Additionally, the IPFP has been shown to produce pro-inflammatory cytokines Prostaglandin E2 (PGE_2_), Interleukin (IL) -1β, and IL-6, which play critical roles in the chemotaxis of immune cells as well as immune cell activation [53,54]. Modeling the synovium and the IPFP will help elucidate the relationship between those two tissues. ASCs harvested from IPFP have been collected from nondiabetic and diabetic total knee arthroplasty and are currently being fully characterized, which will permit us to develop a more specific IPFP adipose tissue for the knee-relevant MPS.

## 5. Conclusions

In this study, we have successfully generated a human stem cell-derived 3D adipose tissue using a custom bioreactor with human ASCs seeded within photo-cross-linked GelMA-based hydrogel. Our findings demonstrate a robust differentiation of ASCs in GelMA into mature adipocytes, as evidenced by the gene expression profile, secreted adipocyte proteins, and the neutral lipid accumulation. These results constitute the baseline characteristics of an engineered adipose tissue model, which will inform our goal of increasing the complexity of the system by the introduction of inflammatory mediators or other tissue and immune components relevant to degenerative pathologies in the knee joint. The successful fabrication of a 3D engineered adipose tissue within a perfusion bioreactor will allow us to perform more complex studies with other tissues and immune cells to ultimately create the functional MPS of an OA knee joint.

## Figures and Tables

**Figure 1 biomolecules-10-01070-f001:**
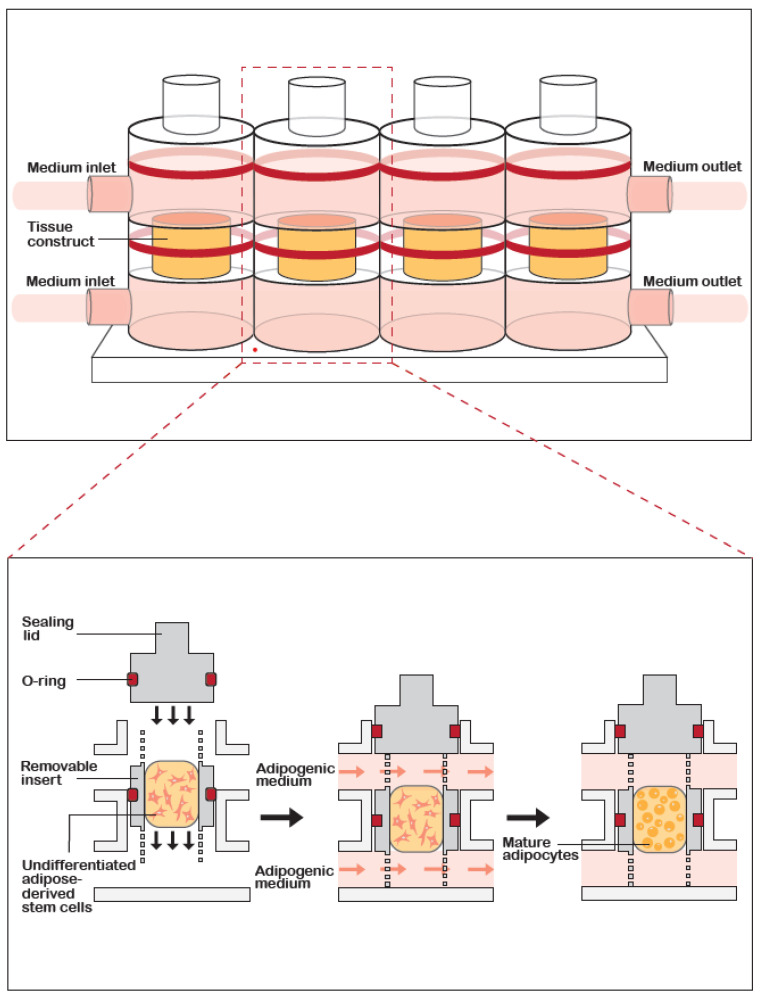
**Schematic of the custom-designed 3D printed bioreactor.** (**A**) Three-dimensional representation of the bioreactor design, showing 4 bioreactors arranged in series. Medium is introduced into inlets both above and below the tissue construct. (**B**) Cross-section view of the bioreactor. Cells are embedded in gelatin methacrylate (GelMA) hydrogel which is first photocured in an insert before being placed into the middle of the bioreactor and the medium is introduced by inflowing into both the top and bottom inlets.

**Figure 2 biomolecules-10-01070-f002:**
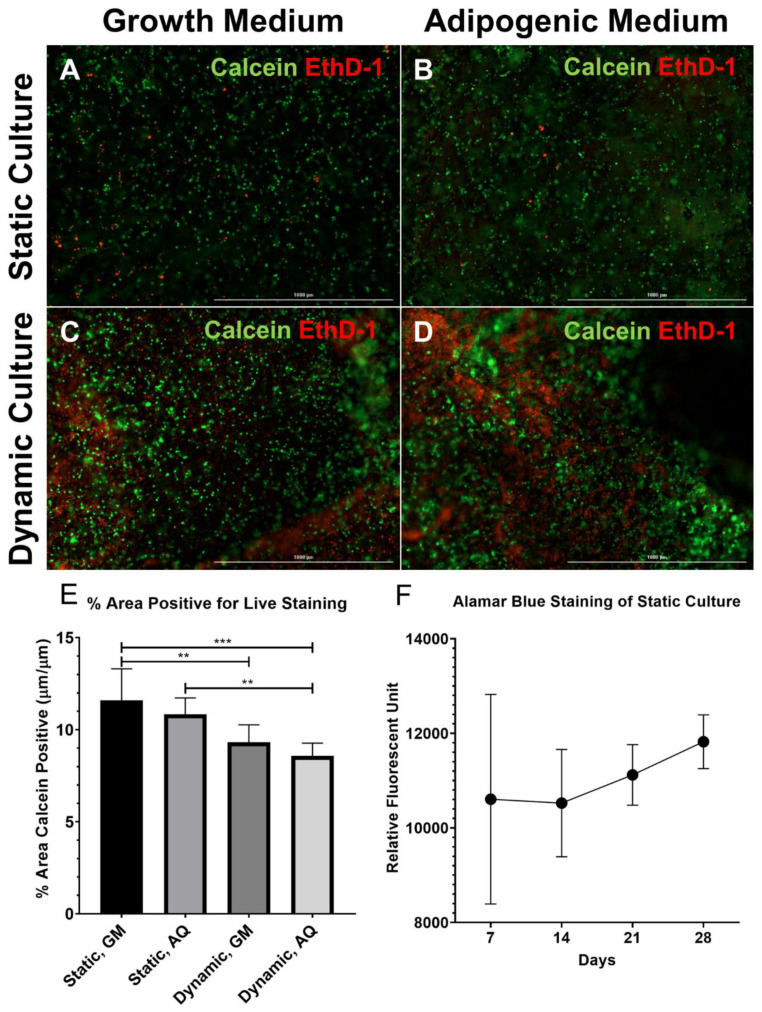
**Robust cell viability in the 3D adipose construct.** (**A**–**D**) Constructs were processed for Live (calcein, green)/Dead (EthD-1, red) staining after 28 days in culture. (**A**,**B**) Static cultures; (**C**,**D**) dynamic cultures (scale bar, 1 mm); (**A**,**C**) growth medium; and (**B**,**D**) adipogenic medium. There was increased EthD-1 positive (dead) staining in the dynamic constructs as compared to the static constructs, while no apparent differences were observed between the cultures in the growth medium versus the adipogenic medium. (**E**) The area positive for Calcein staining over a total area demonstrated a significant decrease in the cell viability in dynamic cultures as compared to static cultures (N = 3, ** *p* < 0.01, *** *p* < 0.001). (**F**) Cell proliferation assessed by the Alamar blue staining of static constructs after 7, 14, 21, and 28 days of culture in growth media (N = 3). Cell proliferation was seen over the four-week culture period. EthD-1, ethidium homodimer-1.

**Figure 3 biomolecules-10-01070-f003:**
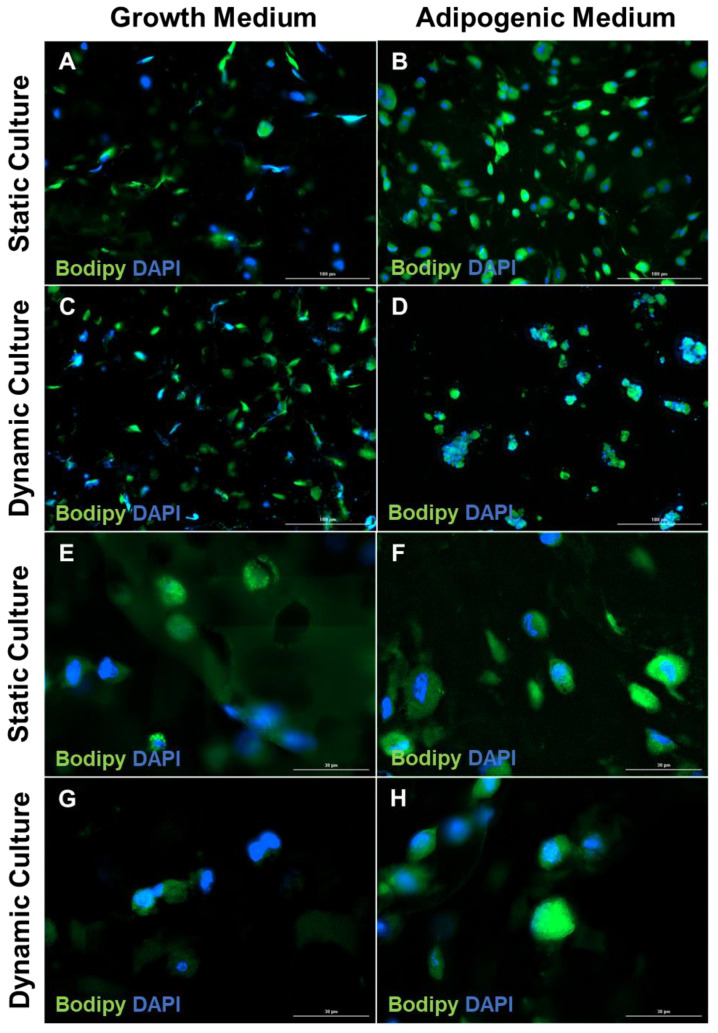
**Lipid accumulation and the loss of the fibroblast phenotype in the cells within the engineered adipose construct.** The constructs were stained with BODIPY and DAPI after 28 days in culture in growth medium or adipogenic medium. (**A**,**B**) Static cultures; (**C**,**D**) dynamic cultures (scale bar, 100 μm). (**E**,**F**) Static cultures; (**G**,**H**) dynamic cultures (scale bar, 30 μm). Both static and dynamic cultures showed an increased accumulation of neutral lipid upon culture in an adipogenic medium. BODIPY, boron-dipyrromethene; DAPI, 4′,6-diamidino-2-phenylindole.

**Figure 4 biomolecules-10-01070-f004:**
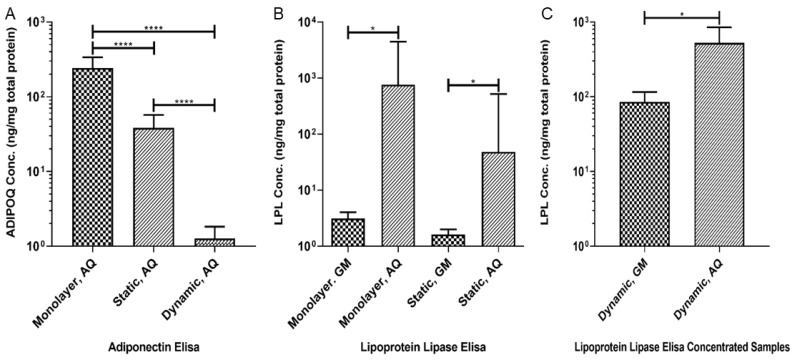
**Increased production of ADIPOQ and LPL in the cultures maintained in the adipogenic medium.** ADIPOQ and LPL were assayed by ELISA after 28 days in culture. (**A**) ADIPOQ concentration was lower in the static and dynamic 3D bioreactor culture when compared with the monolayer culture (N = 4, **** *p* < 0.0001). (**B**) The LPL concentration increased in the cultures maintained in the adipogenic medium and comparably between the monolayer and static 3D culture (N = 4, * *p* < 0.05). (**C**) LPL level increased in the 3D cultures upon maintenance in the adipogenic medium (N = 4, * *p* < 0.05). It should be noted that the culture medium from the 3D dynamic cultures required concentrating to achieve the measurable LPL levels. GM, growth medium; AQ; adipogenic medium.

**Table 1 biomolecules-10-01070-t001:** Gene primers and sequences for RT-PCR.

Name	Forward (5′-3′)	Reverse (5′-3′)
*PPARγ*	AGGCGAGGGCGATCTTG	CCCATCATTAAGGAATTCATGTCATA
*ADIPOQ*	AACATGCCCATTCGCTTTAC	AGAGGCTGACCTTCACATCC
*LEP*	GAAGACCACATCCACACACG	AGCTCAGCCAGACCCATCTA
	AGCACCATAACCTTAGATGGGG	CGTGGAAGTGACGCCTTTCA
*PL1N*	ACAAGTTCAGTGAGGTAG	CCTTGGTTGAGGAGACAG
*LPL*	GAGATTTCTCTGTATGGCACTG	CTGCAAATGAGACACTTTCTC

**Table 2 biomolecules-10-01070-t002:** Gene fold change as determined by qPCR.

Genes	Monolayer		Static		Dynamic	
	Growth Medium	Adipogenic Medium	Growth Medium	Adipogenic Medium	Growth Medium	Adipogenic Medium
*PPAR-γ*	1.06 ± 3.8 × 10^−2^	2.96 × 10^3^ ± 2.90 × 10^3^	1.01 ± 1.0 × 10^−2^	18.58 ± 7.83	1.01 ± 1.0 × 10^−2^	6.91 × 10^1^ ± 3.19 × 10^1^
*APN*	1.09 ± 7.4 × 10^−2^	4.18 × 10^6^ ± 3.62 × 10^6^	1.01 ± 1.0 × 10^−2^	4.45 × 10^4^ ± 3.07 × 10^4^	1.41 ± 4.0 × 10^−1^	2.44 × 10^3^ ± 1.40 × 10^3^
*PL1N*	1.03 ± 4.2 × 10^−2^	7.49 × 10^3^ ± 4.28 × 10^3^	1.07 ± 7.0 × 10^−2^	1.67 × 10^3^ ± 1.42 × 10^3^	1.03 ± 3.0 × 10^−2^	6.70 × 10^1^ ± 3.3.7 × 10^1^
*LPL*	1.48 ± 2.4 × 10^−1^	1.92 × 10^6^ ± 1.86 × 10^6^	1.00 ± 2.53 × 10^−5^	4.67 × 10^3^ ± 4.43 × 10^3^	1.16 ± 1.5 × 10^−1^	5.16 × 10^2^ ± 1.65 × 10^2^
*FABP4*	1.48 ± 4.3 × 10^−1^	2.94 × 10^6^ ± 2.67 × 10^6^	1.02 ± 9.97 × 10^−6^	9.24 × 10^3^ ± 8.97 × 10^3^	1.06 ± 5.0 × 10^−2^	9.01 × 10^3^ ± 5.75 × 10^3^

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
