# Peer review of "Adipose Tissue-Derived Stem Cells Retain Their Adipocyte Differentiation Potential in Three-Dimensional Hydrogels and Bioreactorsâ€"

_biomolecules, 2020, doi:10.3390/biom10071070_

Round 1
Reviewer 1 Report
The manuscript entitled “Adipose Tissue-Derived Stem Cells Retain their Adipocyte Differentiation Potential in 3-Dimensional Hydrogels and Bioreactors” is an interesting study. However, there are some issues that the authors should address.
The introduction needs to be shortened and more focused. The first part of the introduction should be summarized. The authors should better focus the second part of the introduction on what it has been published on IPFP in OA. For example, it has been showed that IPFP of OA patients is not only inflamed and secretes inflammatory molecules but it is also more vascularized and fibrotic compared to controls. Importantly, there is no mention to the recent concept that IPFP may be an anatomo-functional unit with synovial membrane. The third part needs to be shortened. There are several sentences without appropriate references (for example lines 83-86, line 88, lines 98-100 etc.)
Line 78: levles should be corrected.
The authors should report the Alamar blue protocol used in the methods.
The authors reported cell proliferation assessed by Alamar blue staining of static constructs (Fig.2E). However, it is not clear if this graph is related to the static constructs incubated with growth medium or adipogenic medium. Could the authors report all conditions, including those of dynamic cultures (if possible) in order to compare the viability between the different conditions?
Figure 3: the authors should add representative images of cells at higher magnification to better see the morphology and the Lipid accumulation.
Table 2: could the authors report graphs for gene expression analysis and perform also statistical analysis?
Figure 4: this figure can be reported in the supplementary file. Statistical analysis should be performed.
As the authors stated in the discussion, there is a decrease in gene expression of both adipogenic markers and cytokines in both static and dynamic 3D cultures when compared to monolayer, which could be due to diffusion problems. The authors reported that other authors characterized the gelatin and collagen-based scaffolds. However, it is not clear if the constructs used by the authors have been or not characterized. The authors should add the characterization of the constructs.
Given the results obtained, have the authors tried to improve the dynamic constructs changing for example the concentration of GelMA or the thickness/porosity of the constructs?
Line 392-393: The authors reported “this study has demonstrated that the ASCs isolated from fat depots can be used to fabricate an engineered construct that models adipose tissue.” This sentence should be tone down. “Can” should be replaced by “could”.
Line 400: IPTP should be corrected.
Line 399-401: the authors missed the recent concept that IPFP and synovial membrane could act as an anatomo-functional unit. The fabrication of a 3D engineered adipose tissue derived from IPFP in a bioreactor will be useful also to study the cross-talk betwenn these two tissues.
Reviewer 2 Report
The current manuscript addresses the effect of 3D hydrogels and bioreactors in the retaining of MSC adipocyte differentiation. This topic should be considered of importance as a good 3D organ-on-a-chip system in vitro to study the role of the infrapatellar fat pad tissue in the pathobiology of the knee osteoarthritis will help the efforts for a deeper understanding of OA pathologies. The present manuscript is scientifically accurate; authors have described in the introduction section thoroughly the previous literature regarding the existing knowledge on adipose tissue involvement on OA pathology and the efforts to develop microphysiological systems for that, and have presented selected up to date references to cover all these issues. In my opinion the present manuscript is of importance for the reader. However, authors should apply some modifications to improve their manuscript.
Please find below a number of comments that authors should take into consideration:
- Overall introduction section I think its too long and can be significantly shortened without removing valuable information (references etc).
- Introduction, line 130: According to authors the main focus of the manuscript is to investigate how IPFP tissue is related via its secretory profile in the progression of OA. It’s a really nice idea to ex vivo recreate the ‘fat’ fraction of IPFP but the question is why instead of using infrapatellar fat pad-derived MSC (IFP-MSC) that are the physiological residents of the tissue, authors selected subcutaneous adipose tissue-derived MSC (ASCs)? Can authors provide information whatever these two MSC types are identical phenotypically and functionally? Authors have to refer to this matter within the introduction section.
- Materials & Methods, line 169: Why authors selected gelatin as the main substance of their ECM and not collagen that physiologically is the main component of human IPFP ECM?
- Materials & Methods, lines 195-200: Authors have to explain further how they have quantified Live cells after staining (number of fields checked etc) and with which software.
- Materials & Methods, lines 202-219: Authors have to explain further how they have quantified Live cells after staining (number of fields checked etc) and with which software.
- Results/Discussion, lines 326-329: This statement is correct for early-stage differentiated adipocytes. There are previous studies using a defined grading system to characterize the maturation status of adipocytes (doi: 1016/j.jcyt.2012.07.001 and 10.1093/rheumatology/kem217). Authors have to refer to these studies within the text.
- Results/Discussion, lines 344-347: If authors believe that fluid diffusion within the ECM construct was complex what are their observations about cell viability in different levels of the construct (surface and mid part of it)? Did they observe any necrotic core after 28 days? I think authors should add more information/data about this issue also in the Results section.
- Results/Discussion, lines 381-382: Was that the purpose of the current study, as ‘immune components’ were not included?
- Results/Discussion, lines 394-395: Authors should include recent interesting studies on IFP-MSC therapeutic capacity (doi: 1038/s41598-019-47391-2 and 10.1177/0363546520917699)
Round 2
Reviewer 1 Report
A better manuscript after the revision but I have still some comments for the authors.
Figure 2: There are typos that should be corrected by the authors (such as positive and fluercence).
Figure 3: The authors should check the legend and correct the scale bar reported.
The authors moved figure 4 in the supplementary file and reported that no statistical differences were highlighted. What type of statistical test was performed? The authors reported in table 2 that the PPARgamma expression is 2960±2900 in monolayer with adipogenic medium. It is not clear the error bar reported in figure S4 of the same condition. Could the author check?
Line 354: the authors cited fig.S3 but it should be corrected to fig. S4.
The references cited at line 490 about the cross-talk and the anatomo-functional unit between the infrapatellar fat pad and the synovial membrane are not relevant.
